# Genome-Wide Analysis of the UDP-Glycosyltransferase Family Reveals Its Roles in Coumarin Biosynthesis and Abiotic Stress in *Melilotus albus*

**DOI:** 10.3390/ijms221910826

**Published:** 2021-10-06

**Authors:** Zhen Duan, Qi Yan, Fan Wu, Yimeng Wang, Shengsheng Wang, Xifang Zong, Pei Zhou, Jiyu Zhang

**Affiliations:** State Key Laboratory of Grassland Agro-ecosystems, Key Laboratory of Grassland Livestock Industry Innovation, Ministry of Agriculture and Rural Affairs, College of Pastoral Agriculture Science and Technology, Lanzhou University, Lanzhou 730020, China; duanzh12@lzu.edu.cn (Z.D.); yanq16@lzu.edu.cn (Q.Y.); wuf15@lzu.edu.cn (F.W.); wangym19@lzu.edu.cn (Y.W.); wangshsh19@lzu.edu.cn (S.W.); zongxf15@lzu.edu.cn (X.Z.); zhoup2017@lzu.edu.cn (P.Z.)

**Keywords:** *Melilotus albus*, UDP-glycosyltransferases, stress response, coumarin biosynthesis, heterologous expression

## Abstract

Coumarins, natural products abundant in *Melilotus albus*, confer features in response to abiotic stresses, and are mainly present as glycoconjugates. UGTs (UDP-glycosyltransferases) are responsible for glycosylation modification of coumarins. However, information regarding the relationship between coumarin biosynthesis and stress-responsive *UGT*s remains limited. Here, a total of 189 *MaUGT* genes were identified from the *M. albus* genome, which were distributed differentially among its eight chromosomes. According to the phylogenetic relationship, *MaUGT*s can be classified into 13 major groups. Sixteen *MaUGT* genes were differentially expressed between genotypes of Ma46 (low coumarin content) and Ma49 (high coumarin content), suggesting that these genes are likely involved in coumarin biosynthesis. About 73.55% and 66.67% of the *MaUGT* genes were differentially expressed under ABA or abiotic stress in the shoots and roots, respectively. Furthermore, the functions of *MaUGT68* and *MaUGT186*, which were upregulated under stress and potentially involved in coumarin glycosylation, were characterized by heterologous expression in yeast and *Escherichia coli*. These results extend our knowledge of the *UGT* gene family along with *MaUGT* gene functions, and provide valuable findings for future studies on developmental regulation and comprehensive data on *UGT* genes in *M. albus*.

## 1. Introduction

Coumarins, as important secondary metabolites, are widely distributed in the plant kingdom and can be found in many plant roots, flowers, leaves, seeds, and fruits with a variety of biological activities [1]. Simple coumarins, including scopolin, scopoletin, esculin, esculetin, umbelliferone, fraxetin, and sideretin, play diverse roles in the interaction of plants with biotic and abiotic environmental stress factors [2]. In plants, coumarins are present in both their glycosylated (glycoside) and deglycosylated (aglycone) forms [3]. Once synthesized, coumarins are glycosylated and stored in vacuoles [4], as the glycoside form is easier to store and transport. Coumarin affects plant growth and development and is part of the defense response to abiotic stress [5], shapes the root-associated microbial community [6], enhances oxidative stress [7] and nitrate uptake [8], and protects host plants from fungi [9]. To date, coumarins are thought to play certain roles in plant defense due to the induction of their biosynthesis following various stresses [10].

Glycosylation is one of the key modifications in coumarin biosynthesis, and this modification can be used to limit exposure to certain aglycones and promote the solubility, stability, and bioactivity of coumarin for defense and adaptation to environmental changes [3]. The glycosylation of coumarins occurs in the cytoplasm by UDP-glycosyltransferases (UGTs) [11]. All known UGT structures have a GT-B fold, one of the two common glycosyltransferase folds comprised of two N- and C-terminal Rossmann-like domains [12]. In the UGT C-terminal domain, a highly conserved putative secondary product glycosyltransferase (PSPG) box consisting of 44 amino acids can be found, which is involved in the recognition and binding of the sugar donor [11].

UGTs have been previously identified in various plant species, such as *Glycine max* [13], *Gossypium hirsutum* [14], *Cajanus cajan* [15], *Citrus grandis* [16], and *Broussonetia papyrifera* [17]. A total of 122 UGTs have been identified from *Arabidopsis thaliana*, which have been clustered into 14 groups based on their amino acid sequences [18]. Such a high abundance of UGTs in plants demonstrates their indispensable roles in the metabolism of natural products such as secondary metabolites. AtUGT72B1 catalyzes monolignol glucosylation, which is essential for normal cell wall lignifications in Arabidopsis [19]. UGT73A24- and UGT73A25-catalysed phytoalexin N-feruloyl tyramine glucosylation may strengthen cell walls to prevent the intrusion of pathogens in *Nicotiana benthamiana* [20]. UGT73F18 and UGT73F19 mediate hemolytic sapogenin modification in *Medicago truncatula* [21]. AtUGT76F1 strongly favors IPyA as a substrate and antagonizes YUCs to regulate auxin biosynthesis and auxin responses in Arabidopsis [22]. The glycosylation activity of UGT73C7 results in the redirection of phenylpropanoid metabolic flux to the biosynthesis of hydroxycinnamic acids and coumarins, promoting *SNC1*-dependent Arabidopsis immunity [23].

As a great potential forage source for ruminants, *Melilotus albus* is a rich source of bioactive coumarins [24]. It has been used for both forage production and soil improvement [25]. This plant demonstrates high tolerance in extreme environments, such as drought, cold, and high salinity [26]. Glycosylation plays an important role in enhancing plant resistance to abiotic stress [15,27]. However, the functional redundancy of *UGT* genes and complex glycosylation modifications of coumarin make it difficult to functionally characterize all *UGT* genes of *M. albus*. In this study, we aimed to identify putative *UGT* genes involved in coumarin biosynthesis and abiotic stress and provide potential candidate *MaUGT* genes for coumarin engineering and further functional analysis and application in plants. Herein, using the latest available genome assembly and annotation database, we identified all 189 *UGT* genes in the *M. albus* genome and found that they were differentially expressed in various tissues. We further functionally characterized two *MaUGT* genes that are potentially involved in coumarin biosynthesis in vitro. Our results demonstrated that the genes *MaUGT68* and *MaUGT186* can respond to abiotic stressors and that *MaUGT186* is responsible for the glycosylation of scopoletin, which will lead to more in-depth investigation of the function of *UGT* genes in *M. albus*.

## 2. Results

### 2.1. Genome-Wide Identification and Characteristics of the UGT Genes in *M. albus*

After a BLSTP search, we initially obtained 205 candidate genes. All putative genes were subsequently verified in PFAM databases to confirm the existence and the number of the PSPG domain. A total of 189 *UGT* genes were identified in the *M. albus* genome. Their physical locations were assigned differentially to the eight chromosomes, and relatively high densities were observed on chromosomes 3, 5, 6, 7, and 8 (Appendix A). In particular, several *MaUGT* genes were clustered at a certain location on these chromosomes (Appendix A), suggesting that a gene duplication event may have occurred during evolution. The 189 MaUGT proteins corresponding to the genes were named according to their physical location (from top to bottom) on chromosomes 1–8. As shown in Appendix A, the *MaUGT* gene encoded proteins ranging from 134 aa (14.54 kDa) to 969 aa (109.59 kDa). Their isoelectric points (pIs) ranged from 4.86 to 8.83, and the grand average of hydrophilicity (GRAVY) ranged from −0.444 to 0.144. The predicted subcellular localizations of the MaUGT proteins revealed that 137 UGT members (72%) were probably in the cytoplasm, and 42 (22%) and nine (5%) UGTs were most likely in the plasma membrane and chloroplast, respectively. Only two UGTs were predicted to be located in the nucleus.

### 2.2. Phylogenetic Analysis of MaUGTs

In previous studies, Arabidopsis UGT was first used to study evolution. Its UGT phylogenetic tree was divided into 14 different groups (A–N) [18]. Later, three other groups (O, P, R) were discovered during the study of *M. truncatula* glycosyltransferases [28]. In this study, 38 UGTs, including 19 *Arabidopsis* UGTs and 19 *M. truncatula* UGTs, were downloaded as references with 189 MaUGTs to construct a phylogenetic tree (Figure 1). Analysis of the phylogenetic tree revealed that all *UGT* members were divided into 17 phylogenetic groups, including 14 conservative groups (A–N) identified in *A. thaliana* and three additional identified groups (O, P, and R) found in *M. truncatula* and *M. albus*. The 189 *UGT*s from *M. albus* were unevenly clustered into 13 groups. Five phylogenetic groups, A, D, E, G, and L, seem to have expanded to a greater extent than the others during higher plant evolution. In *M. albus*, these five phylogenetic groups expanded more than the other groups. There were 52 (27.5%), 39 (20.6%), 33 (17.5%), 24 (12.7%), and 21 (11.0%) members in Groups E, D, G, L, and A, respectively. Both the H and O groups contained nine members. Groups B, I, J, M, N, and P had three, one, two, two, one, and two *MaUGT*s, respectively. However, Groups R, C, K and F did not have any members from *M. albus*.

### 2.3. Structure, Motif Distribution Analysis, and Domain Architecture of the MaUGT Genes

To gain insights into the structural and functional characteristics of the *UGT* gene family in *M. albus*, an analysis of the gene structure of 189 *MaUGT* genes in terms of intron and exon organization was conducted. The analysis suggested that *MaUGT* genes contained one–three exons with varying numbers of introns (Appendix A). In parallel with the phylogenetic relationship in Figure 1, the closely associated *MaUGT* gene clustering within the same group displayed close similarity in gene structure with respect to intron number and exon length. Among the 189 UGTs, 113 had no introns, and 63 contained one intron each (Appendix A). For the remaining 13 UGTs, eleven had two introns, and two had three introns. For the UGT groups, the largest number of genes losing introns was observed for Group E, with 45 members, followed by 33 in Group D and 18 in Group A. A total of 27 (81.8%) UGTs in Group G contained one intron, followed by 12 in Group L (Appendix A).

A total of 76 *MaUGT*s containing intron sequences were observed. After mapping the introns to the amino acid sequence alignment, at least 10 independent intron insertion events were observed. These insertion events are serially numbered I-1 to I-10 according to their position (Figure 2). Highly conserved introns were observed for I-4 (intron 4), containing 45 *MaUGT*s belonging to Groups G, H, I, N, J, and P. Among these groups, all members of Groups H, I, N, J, and P contained intron 4. In Group G, 28 of 33 *UGT*s had the highly conserved intron 4. Intron 5 was predominantly observed in Groups D and L.

Among the 91 introns detected in *MaUGT* sequences, 26, 58, and seven were in phases 0, 1, and 2, respectively (Figure 2). For the highly conserved intron 4, only two were in phase 0, one was in phase 2, and phase 1 accounted for 93% of all introns (Figure 2). For intron 5, 16 out of 17 *UGT*s were in phase 0. This suggests that the highly conserved introns were in the same intron phase. Taken together, these results indicated that *UGT* genes placed within different groups are usually distinct, with each group showing its common gene structural arrangement.

To further gain an understanding of structural characteristics, putative motifs were predicted using MEME Suite, and 15 distinct motifs were identified. Appendix A shows that groups classified by phylogenetic analysis shared similar conserved motif compositions. Motif 1 contained the complete PSPG-box motif, and all members included this motif; motif 11 was only found in Groups G and J, with just 23 members; motif 8 was found in 37 members located in Groups D, L, G, M, and I; motif 10 was found in 41 members, and most of these were concentrated in Groups G and L. To some extent, these specific motifs may lead to functional differences in the *UGT* genes in *M. albus.*

In addition, Appendix A shows the 44 amino acids that comprise the *MaUGT* family PSPG-box motif. Positions 1 (W), 4 (Q), 8 (L), 10 (H), 19–24 (HGWNS), 27 (E), 32–34 (GVP), 39 (P), 43 (D/E), and 44 (Q) were found to be highly conserved amino acids, most of which are related to sugar donor binding [11].

### 2.4. Characterization of Cis-Acting Elements in the MaUGT Gene Promoters

To understand the transcriptional regulation of *MaUGT* genes, the upstream promoter regions (2.0 kb in size) of *MaUGT* genes were analyzed for the prediction of potential *cis*-acting elements using the PlantCARE tool. A total of 20 *cis*-acting elements were recorded in this study (Appendix A), and these identified *cis*-acting elements could be classified into four functional categories: light response, stress response, hormone response, and developmental regulation. As a result, a total of 531 Box 4 *cis*-acting elements related to light response were identified, which was more than the number of other *cis*-acting elements. Moreover, we found at least three *cis*-acting elements in each *MaUGT* gene promoter, and it is quite possible that *MaUGT* genes are more likely to respond to multiple abiotic stresses. These results suggest that the *MaUGT* genes in *M. albus* play a vital part in the complex hormone regulatory network and may be involved in a variety of stress responses and the synthesis of secondary metabolites.

### 2.5. Transcriptome Analysis of Tissue-Specific Expression

Different tissues of plants usually contain different secondary metabolites, and diversified glycosyltransferases also have multiple functions for specific metabolites [15]. The RNA-seq data from *M. albus* tissues showed that the *UGT*s were variably expressed in all the investigated tissues (root, stem, leaf, flower, and seed) (Appendix A). Out of the 189 genes, 149 genes displayed expression values (FPKM) ≥1 in at least one of the tissues under comparison. Among these genes, 33, 41, 18, 40, and 31 *MaUGT* genes were preferentially expressed in leaves, roots, stems, flowers, and seeds, respectively (Appendix A), suggesting that they may function in different tissues for the glycosylation process. Each *UGT* gene showed a unique expression pattern and no expression patterns were linked with the phylogenetic groups.

### 2.6. Expression Analysis of Coumarin Biosynthesis-Related MaUGT Genes

To select the *MaUGT* genes associated with coumarin biosynthesis, the leaves in the flower stage of four near-isogenic lines (NILs) of *M. albus* (Ma46, Ma47, Ma48, and Ma49) and the recurrent male parent (RP) were used for RNA-seq. Among these, the coumarin contents in Ma48, Ma49, and RP were significantly higher than those in Ma46 and Ma47 [29]. The heatmap shows that the genotypes clustered together according to their coumarin levels, and the *M. albus* genotypes with the same coumarin levels represented similar gene expression patterns (Appendix A), indicating that coumarin levels should be dynamic due to changing expression of *MaUGT* gene regulation. Comparisons of gene expression between Ma46 and Ma49 showed 16 differentially expressed *MaUGT* genes. The number of differentially expressed *MaUGT* genes was greater than that in Ma47 vs. Ma48, which showed nine differentially expressed *MaUGT* genes (Figure 3a). In total, 11 and 12 *MaUGT* genes were significantly differentially expressed between Ma46 vs. RP and between Ma47 vs. RP, respectively. The numbers of upregulated and downregulated *MaUGT* genes are shown in Figure 3b. We also found four upregulated genes and one downregulated gene that overlapped between Ma46 vs. Ma49 and Ma47 vs. Ma48, respectively (Figure 3b). These genes are likely associated with coumarin biosynthesis. Furthermore, six *MaUGT* genes, which were significantly differentially expressed between Ma46 vs. Ma49, were selected to verify the expression pattern in Ma46 and Ma49 by qRT–PCR (Figure 3c). The expression patterns of the selected genes showed significant expression changes between Ma46 and Ma49.

### 2.7. Expression Analysis of ABA and Abiotic Stress-Related MaUGT Genes

To investigate the expression levels of *MaUGT* genes under ABA and abiotic stresses, we analyzed the gene expression in the shoots and roots of Ma46 under ABA, drought, and salt stress conditions using RNA-seq data. As a result, high expression of *MaUGT*s was observed in roots in comparison to the shoots. A total of 189 (100%) and 132 (69.84%) *MaUGT* genes showed gene expression under at least one stress condition (FPKM ≥ 1) in the roots and shoots, respectively (Appendix A). Furthermore, 126 (66.67%) and 139 (73.55%) were differentially expressed |log_2_(fold change) ≥ 1| under at least one stress treatment in the roots and shoots, respectively. Eighty-seven, 64, and 86 differentially expressed *MaUGT* genes were identified in *M. albus* roots under ABA, drought, and salt stress, respectively. Compared to the controls, there were 69, 102, and 47 differentially expressed *MaUGT* genes in shoots under ABA, drought, and stress treatments, respectively (Figure 4). Interestingly, 34 and 20 of these genes overlapped under the three stress conditions in the roots and shoots, respectively. A total of 111 *MaUGT* genes were differentially expressed in both the roots and shoots under stress conditions. These genes were mainly distributed in Groups E, D, A, G, and L. In Group E, 33 of 52 genes were differentially expressed in the roots and shoots under stress conditions. The ABA and drought treatments had the most overlapping genes (13 in roots; 35 in shoots), whereas the ABA and salt stress treatments had the fewest overlapping genes (26 in roots; one in shoots), indicating that more ABA-independent *MaUGT* genes were involved in the response to abiotic stress.

Additionally, the *MaUGT* genes potential associated with coumarin biosynthesis and responding to ABA, drought, and salt stresses were selected for qRT–PCR analysis to verify their expression patterns in response to exogenous ABA (100 mM ABA; 24 h), drought (20% PEG-6000; 24 h), and salt (250 mM NaCl; 24 h) treatment. Based on these analyses, we found that the qRT–PCR data had similar expression trends as those shown in the RNA-seq data. In addition, the change trends in *MaUGT* gene expression in the shoots and roots were consistent (Figure 5). For example, *MaUGT68* and *MaUGT186* were upregulated in shoots and roots under drought and salt stresses.

### 2.8. MaUGT68 and MaUGT186 Improve Drought Tolerance in Yeast

*MaUGT68* and *MaUGT186* were differentially expressed between Ma46 and Ma49, and they were significantly upregulated under abiotic stresses and the trends of FPKM and qRT-PCR were consistent. Thus, we hypothesized that as two representative genes they play an invaluable role in *M. albus* to cope with coumarin biosynthesis and abiotic stress. The effects of *MaUGT68* and *MaUGT186* on yeast growth and stress (drought and salt) resistance were analyzed in yeast cells harboring the pYES2-*MaUGT68* and pYES2-*MaUGT186* constructs. The results showed no difference among transformed lines and an empty vector line under normal culture conditions. The growth of transformed yeast was not significantly affected by 5 M NaCl treatments, but the MaUGT68 transformed yeast cells still grew under 10^5^-fold dilution. However, we observed that the transformed yeast cells exhibited prominent resistance under 30% PEG-6000 treatments, especially under 10^5^-fold dilution (Figure 6).

### 2.9. MaUGT186 Is Involved in Scopolin Biosynthesis

We thus heterologously expressed *MaUGT68* and *MaUGT186* in *Escherichia coli* to explore the ability of the recombinant proteins to glucosylate scopoletin. SDS–PAGE analysis showed that two protein products were observed for *MaUGT68* and *MaUGT186* with relative molecular masses of 53.17 and 52.03 kDa, respectively (Figure 7a). High-performance liquid chromatography (HPLC) analysis of the extracts of incubation mixture of UDP-glucose and scopoletin as substrates, together with either the UGT68 or UGT186 protein, showed that scopoletin was only glucosylated by MaUGT186, producing potential scopolin with exactly the same retention times as the authentic standards of scopolin. These results implied that MaUGT186 was likely to be associated with the glucosylation of scopoletin in *M. albus*.

## 3. Discussion

UGTs have important roles in growth, development, and interactions with the environment. They play an essential role in regulating glucose metabolism and homeostasis, and they are also important for the biosynthesis, storage, and transport of secondary metabolites [30]. Although *UGT*s have been identified in many different species, the identification and molecular characterization of this gene family in *M. albus* have not been previously reported. In our study, 189 *UGT* genes were identified. Compared to other plants, the results suggested that the *UGT* family did not exhibit significant expansion in *M. albus*. For example, 145 *UGT* genes were found in *Citrus grandis* fruits [16]. A total of 208, 94, and 243 *UGT* genes were identified in *G.*
*max, Lotus japonicus* and *M. truncatula*, respectively [28].

Evolutionary analysis revealed that 189 *UGT* genes from *M. albus* could be classified into 13 groups. Compared to the 17 reported groups (A–Q) [31], Groups C, K, F, and R were missing in *M. albus* (Figure 1). Our study also indicated that *M. albus* Group E was a large group consisting of eighteen 71-family *UGT*s, twenty-eight 72-family *UGT*s, and six 88-family *UGT*s (Appendix A), accounting for 27.51% of the *UGT*s in *M. albus*. This group appears to have expanded to a greater extent than any of the other groups during the evolution of higher plants and possesses similar activities to those from Group D [32]. In addition, many plant *UGT* gene members belonging to Groups E and D have been functionally identified, including the glycosylation of benzoates [33], flavonoids [34], brassinosteroids [35], terpenoids [36], lignin [19], hydroxycinnamic acids (HCAs), and coumarins [23], which indicates that Group E makes an important contribution to the glycosylation of plant secondary metabolites. Therefore, the *MaUGT*s in Groups D and E are more likely to be responsible for coumarin glycosylation modification.

Intron gain and loss events, in addition to the positions and phases of introns relative to protein sequences, are important clues to understand evolution [29]. Intron mapping of the 189 *MaUGT*s revealed that 59.8% of the members lacked introns, which is more than the number (43%) of *Prunus persica* [30] and the number (48%) of *C. grandis* [16] *UGT* genes but close to the number (60%) of those in *Zea mays* [31] and the number (58%) in Arabidopsis [18]. Ten intron positions were identified in *MaUGT* genes, with I-4 (intron 4) being the most widespread intron (Figure 2). Intron 4 was observed in most members of Groups G–J, N, and P and was regarded as the oldest in *MaUGT*s. The second highly conserved intron was intron 5, which was observed in most members of Groups D and L in *M. albus*. It is worth noting that, among intron 4, many were in phase 1, suggesting that the majority of conserved introns were ancient elements and that their phases remained stable because deletion or insertion of small DNA fragments that cause a phase change may lead to changes in gene function and be eliminated by natural selection [37]. Structural investigation of the PSPG motif in each phylogenetic group revealed the role of specific amino acid residues that are highly conserved at positions 1 (W), 4 (Q), 8 (L), 10 (H), 19–24 (HGWNS), 27 (E), 32–34 (GVP), 39 (P), 43 (D/E), and 44 (Q) (Appendix A). The last glutamine (Q) residue within the PSPG motif is thought to confer specificity for UDP-glucose as the sugar donor [38]. The occurrence of these specific amino acids at these positions in the sequence provides certain evolutionary and functional information that may be helpful for enzyme discovery [32].

The analysis of promoter regions suggested that some of the *UGT* genes contain a secondary metabolite and stress-related element, including MBSI, WUN-motif, LTR, MBS, and TC-rich repeat *cis*-acting elements, and the majority of *MaUGT* genes belonging to different groups were found to contain a significant number of MYB binding sites for some important classes of plant transcription factor genes. Many studies have strongly demonstrated that MYB transcription factors function as key regulators of plant secondary metabolism, such as flavonoid and coumarin biosynthesis [6,39,40], implying that some *MaUGT* genes may play a significant role in the synthesis of flavonoids and coumarins.

Plants endure many environmental stresses during their lifespan and have evolved the ability to deal with harsh environmental clues. Previous studies have shown that some *UGT* genes are involved in the response to multiple stresses. Overexpression of *UGT76E11* increases flavonoid accumulation and enhances abiotic stress tolerance in Arabidopsis [41]. *SlUGT75C1* plays a crucial role in ABA-mediated fruit ripening, seed germination, and drought response in tomato [42]. Overexpression of *CrUGT87A1* increases salt tolerance by accumulating flavonoids for antioxidation in Arabidopsis [43]. Here, 66.67% and 73.55% ABA, drought, and salt stress-related *MaUGT* genes were identified in the roots and shoots, respectively (Appendix A). Thirty-four and 20 genes showed differential expression under the three stress conditions in the roots and shoots, respectively (Figure 4). These results indicate that *MaUGT* genes are involved in the regulation of abiotic stresses and ABA-dependent pathways.

We further found that seven common *MaUGT* genes showed differential expression between the NILs and the differential expression in stress (Appendix A), revealing that these genes may involve in coumarin biosynthesis and abiotic stress. Furthermore, the six selected differentially expressed genes between Ma46 and Ma49 also showed significant expression changes under abiotic stress by qRT-PCR verification (Figure 3 and Figure 5). These results show that coumarin content may be affected by environmental stresses. There is consistent evidence that coumarin improves plant tolerance to salinity by enhancing antioxidant defense, the glyoxalase system, and ion homeostasis [44]. Here, yeast expression showed that *MaUGT68*- and *MaUGT186*-transformed yeast gained improved tolerance to drought stress. However, with regard to the salt response, the *MaUGT68*- and *MaUGT186*-transformed cultures showed no obvious salt resistance (Figure 6), indicating the difference of *MaUGT* genes in stress tolerance. Although both *MaUGT68* and *MaUGT186* were mainly induced by drought stress, they were not necessarily empowered with similar salt response abilities, which may be correlated to multiple and complex regulation pathways involved in the stress response. The enzymatic characteristics of MaUGT68 and MaUGT186 showed that the MaUGT186 protein can use scopoletin as a substrate; however, MaUGT68 did not exhibit any activity towards scopoletin (Figure 7b), suggesting that their function may have differentiated during evolution. This is consistent with the suggestion that the function and substrate specificity of UGT enzymes are not predictable based on sequence analysis alone [45].

## 4. Materials and Methods

### 4.1. Identification of the UGT Genes in *M. albus*

To recognize all putative *UGT* genes of *M. albus*, a local BLAST search using the *M. albus* genome was performed. The protein sequences of 122 UGTs from *A. thaliana* and 243 UGTs from *M*. *truncatula* retrieved from Phytozome v12.1 (https://phytozome.jgi.doe.gov/, accessed on 14 May 2021) were used as queries in BLASTP searches to identify all UGTs from *M.*
*albus* (*e*-value < 10^−5^). Then, the conserved domains of all possible protein sequences were further confirmed by PFAM (http://xfam.org/, accessed on 14 May 2021). Next, the cDNA sequences, amino acid sequences, and genomic sequences were extracted for further analysis. The molecular weights (MWs), theoretical isoelectric points (pIs), and grand average hydropathicity (GRAVY) values of the MaUTG proteins were calculated with the online program ExPASy (http://web.expasy.org/protparam/, accessed on 25 May 2021) [46]. Subcellular localization of UGT proteins was predicted using the online analysis tool CELLO v2.5 system (http://cello.life.nctu.edu.tw, accessed on 25 May 2021) from the Molecular Bioinformatics Center [47]. The chromosome distribution of the *UGT* genes was conducted with TBtools software [48].

### 4.2. Phylogenetic and Classification Analysis

The alignments of the full-length amino acid sequences of 189 MaUGTs, 19 AtUGTs, and 19 MtUGTs were performed using Clustal X with the default settings [49], and the results were displayed by DNAMAN software. An unrooted phylogenetic tree was constructed based on the alignments using MEGA 7.0 with the neighbor-joining (NJ) method, and bootstrap tests were performed using 1000 replicates to support statistical reliability [50].

### 4.3. Analysis of Conserved Motifs, Gene Structure, and Exon/Intron Organization of the MaUGTs

The deduced amino acid sequences of the MaUGTs were analyzed by the MEME program with a maximum number of motifs of 15 (http://meme-suite.org/, accessed on 17 May 2021) for conserved motif analysis [51]. The amino acid sequences in the MaUGT motifs were manually adjusted using the Weblogo online program (http://weblog.berkeley.edu/logo.cgi, accessed on 17 May 2021) to display the characteristics of the PSPG box [52]. The gene structure and intron phases were graphically displayed by the Gene Structure Display Server (http://gsds.cbi.pku.edu.cn/, accessed on 21 May 2021) based on the genomic sequences and the corresponding coding sequences (CDSs) [53]. The intron can be inserted anywhere in the transcript, they are located either between codons or within codons, which are termed intron phases, and the phases and the locations usually stay unchanged for a long time. Specifically, intron phases were determined as follows: introns positioned between two codons were defined as phase 0, introns positioned between first and second base of codon were defined as phase 1, and introns positioned between second and third base were defined as phase 2. After mapping the introns to the amino acid sequence alignment, 10 independent intron insertion events were observed according to their position [16].

### 4.4. Analysis of Promoter Cis-Acting Elements

Sequences of 2.0 kb from the promoters of the *MaUGT* genes were used to identify the *cis*-acting elements in the promoter regions through the PlantCARE website (http://bioinformatics.psb.ugent.be/webtools/plantcare/html/, accessed on 6 June 2021) [54].

### 4.5. Transcriptome Analysis for Tissue-Specific Expression

Root, stem, and flower samples were collected during the flowering stage. Seed samples were mixed from unmatured, imbibed, and germinated seeds at 2 days. Leaf samples of four NILs of *M. albus* and the recurrent male parent of RP were collected during flowering. The methods of RNA-seq and data analysis were the same as those in a previous study [28,55]. Quantification of gene expression levels were estimated by fragments per kilobase of transcript per million fragments (FPKM) mapped using StringTie (http://ccb.jhu.edu/software/stringtie/, accessed on 25 May 2020) in each sample [55].

### 4.6. Stress Treatment and Transcriptomic Data Analysis

Ma46 genotype plants were grown in a glasshouse under controlled conditions. For drought stress, six week old plants were treated with 20% PEG-6000. The shoots and roots were collected at 0, 3, and 24 h after drought treatment. For ABA treatment, six week old plants were treated with 100 mM ABA. The shoots and roots were collected at 0, 1, and 24 h after ABA treatment. All samples were immediately frozen in liquid nitrogen and stored at −80 °C. Three replicates were performed for each sample. The methods of RNA-seq and data analysis were the same as those in a previous study [28]. Transcriptome data of the *MaUGT* genes under salt stress were obtained from our previous study [55].

### 4.7. Quantitative Real-Time PCR Analysis

Total RNA was extracted from the shoots (leaves and stems) and roots of control and treated seedlings using TransZol reagent (TransGen Biotech, Beijing, China). Reverse transcription and cDNA synthesis were performed using a TIANScript Ⅱ RT Kit (Tiangen, Beijing, China) from 1 μg of total RNA. qRT–PCR was performed on each cDNA template using Hieff^®^ qPCR SYBR^®^ Green Master Mix (No Rox) (Yeasen Biotech Co., Ltd., Shanghai, China) on a CFX96 Real-Time PCR Detection System (Bio–Rad, Los Angeles, CA, USA). The *tubulin* gene was used as an internal control. Primer sets are listed in Appendix A. The expression levels were calculated relative to the controls and determined using the 2^−ΔΔCT^ method [56]. Each of the three biological replicates was supported by three technical replicates.

### 4.8. Validation of Heterologous Expression in Yeast

To produce the pYES2-*MaUGT68* and pYES2-*MaUGT186* constructs, the full-length coding sequences of *MaUGT68* and *MaUGT186* were amplified from Ma46 and cloned into the pYES2 expression vector using a ClonExpress^®^ MultiS One Step Cloning Kit (Vazyme Biotech Co., Ltd., Nanjing, China) following the manufacturer’s protocol with the specific primers listed in Appendix A. After confirmation of sequences, the recombinant pYES2-*MaUGT68* and pYES2-*MaUGT186* plasmids and empty pYES2 plasmid were transformed into *Saccharomyces cerevisiae* strain INVSc1. The two yeast cultures were independently grown in synthetic complete (SC)-Ura liquid medium containing 2% (*m*/*v*) galactose at 30 °C for 36 h up to A_600_ = 0.4. Then, the yeast was collected and adjusted with SC-Ura including 2% galactose and cultivated up to A_600_ = 1 for stress analysis. The same number of cells was resuspended in 30% PEG-6000 and 5 M NaCl [57]. The treated yeast liquid was diluted to 1:10 and cultured on SC-U/2% (*w*/*v*) glucose agar plates for 2–3 days to observe colony growth, and photos were taken to record the expression of the binding protein.

### 4.9. Heterologous Expression in E. coli and Glucosyltransferase Activity Assays

The full-length cDNA sequences of *MaUGT68* and *MaUGT186* were cloned into the pET32a vector between the BamH I and Sac I sites. The constructs were then introduced into *E. coli* strain BL21(DE3), and the transformants were grown in LB medium containing 100 mg/mL ampicillin at 37 °C until OD_600_ reached 0.6. Expression of the MaUGT68 and MaUGT186 proteins were induced by addition of 0.1 mM isopropyl β-D-1-thiogalactopyranoside and incubation at 28 °C for 18 h. The method of heterologous expression in *E. coli* was described in a previous study [58]. The expressed proteins were confirmed using 12% SDS–PAGE.

BL21(DE3) cells harboring the pET32a-*MaUGT68* and pET32a-*MaUGT186* vectors and empty vector were cultured in LB medium containing 100 mg/mL ampicillin at 37 °C until OD_600_ reached 0.6. After IPTG was added for 9 h, the substrate scopoletin (final concentration of 100 μM) and UDP-glucose (final concentration of 500 μM) were added, and culture was continued for 9 h. Then, the reaction system was extracted with an equal amount of ethyl acetate. The recovered ethyl acetate was concentrated and dried and then the residue was dissolved in MeOH for HPLC analysis.

The HPLC separation was performed with an Agilent 1100 HPLC system using a 5 μm C18 column (4.6 mm × 150 mm, Agilent-XDB), maintained at 30 °C, with water (contained 0.1% phosphorous acid) and acetonitrile as mobile phase. The flow rate of the mobile phase was set at 1 mL/min over 20 min, and scopletin and scopolin were monitored by retention time in comparison with authentic standards at 346 nm absorbance.

## 5. Conclusions

In summary, this study represents the first comprehensive identification and analysis of the *MaUGT* family in *M. albus*. A total of 189 *UGT* genes in *M. albus* were identified, and their phylogenetics, chromosomal location, gene structure, and structural domains were analyzed to reveal the rules of the *UGT* family in *M. albus*. The expression analysis in this study provides a global landscape of the expression of *MaUGT*s in different tissues. The expression analysis of *UGT*s provides candidates for further functional studies of *MaUGT* genes during stress response and coumarin biosynthesis regulation. Our study provides systematic insights into the potential roles of *UGT*s in *M. albus*, which is helpful for screening candidate genes and studying the functions of *MaUGT* genes, but a series of experiments are still required in the future to confirm their functions. 

## Figures and Tables

**Figure 1 ijms-22-10826-f001:**
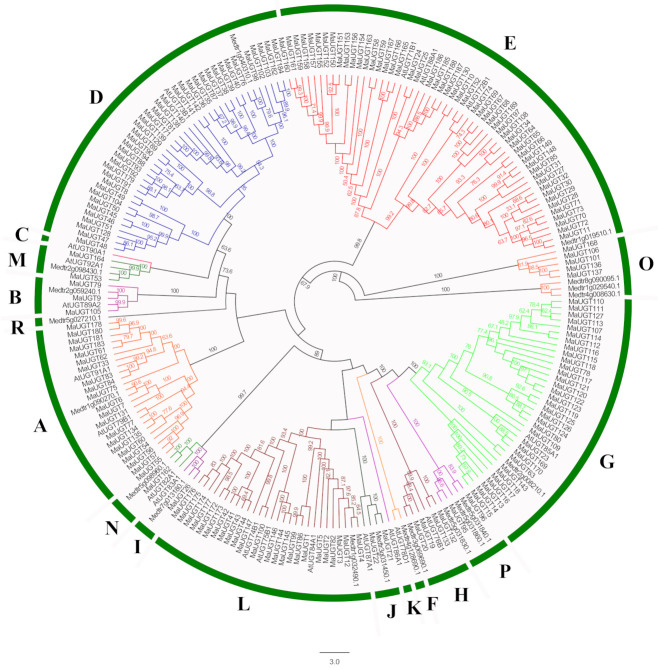
Phylogenetic analysis of the UGTs from *M. albus*. A total of 189 UGT proteins from *M. albus*, 19 UGTs from Arabidopsis, and 19 UGTs from *M. truncatula* were used to construct the neighbor-joining tree using the program MEGA 7.0. All protein sequences were full length, and the bootstrap values of 1000 replicates were calculated at each node.

**Figure 2 ijms-22-10826-f002:**
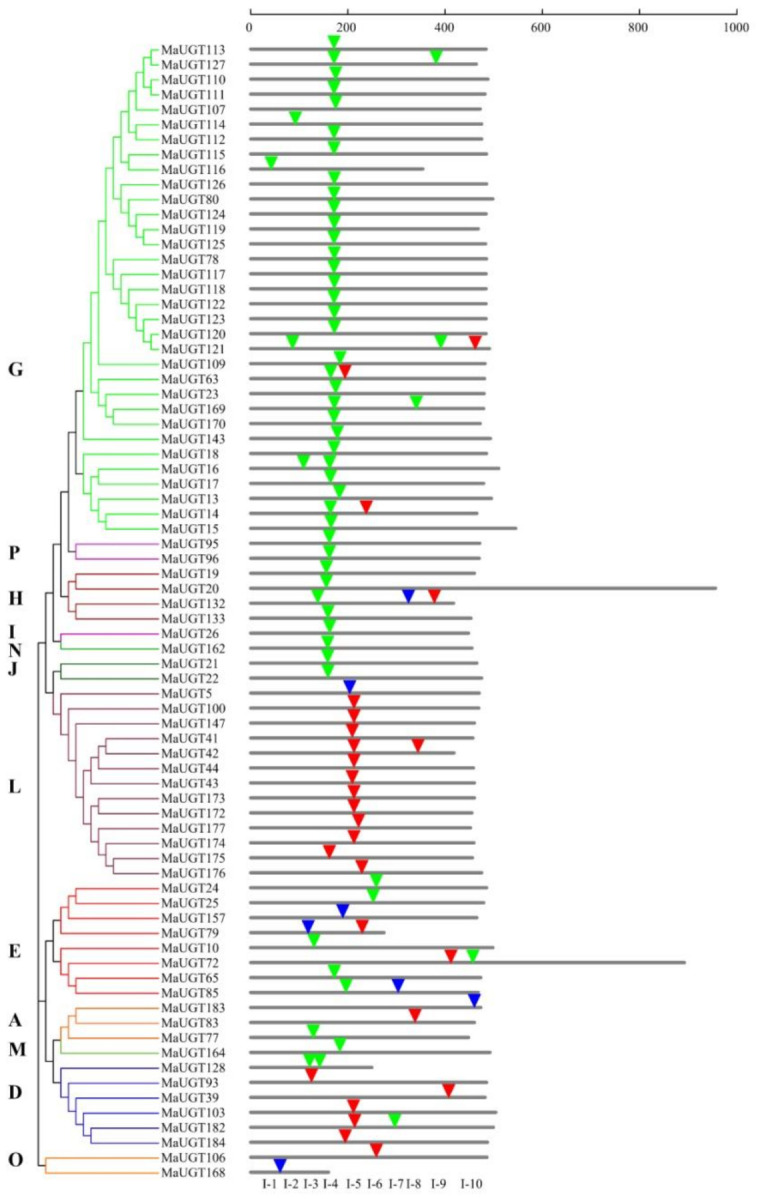
Intron distribution among 76 *MaUGT* genes. The introns were mapped and numbered to the alignment of the amino acid sequences encoded by the *UGT* genes. The scale on the top of the map shows the intron insertion on each gene. Grey thick lines indicate the length of the amino acid sequences. Inverted triangles indicate the positions of introns that occurred on each gene. Intron phases are indicated by red triangles, green triangles, and blue triangles for 0, 1, and 2, respectively. The classification of *MaUGT* genes is indicated by the phylogenetic relationship on the left, and different phylogenetic groups are distinguished by different colors.

**Figure 3 ijms-22-10826-f003:**
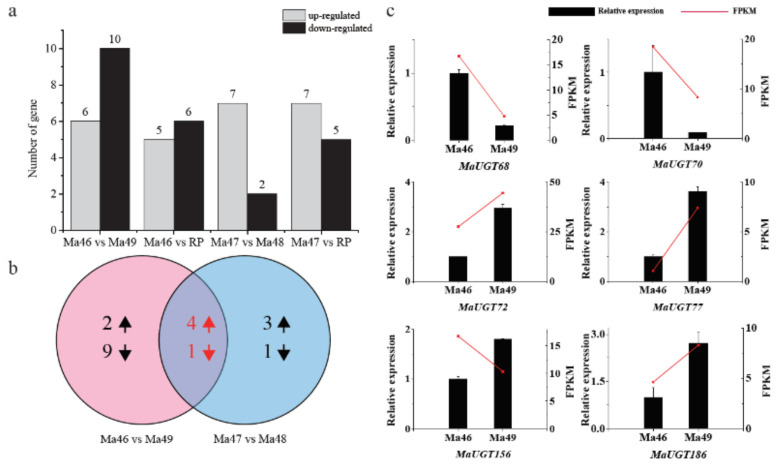
Differential gene expressions of coumarin biosynthesis-related *M**aUGTs*. (**a**) The number of up and downregulated genes in the comparisons of Ma46 vs. Ma49; Ma46 vs. RP; Ma47 vs. Ma48; and Ma47 vs. RP. The number at the top of the column represents the differentially expressed *MaUGT* gene numbers. (**b**) Venn diagrams of the DEGs from Ma46 vs. Ma49 and Ma47 vs. Ma48. Up arrows represent upregulation, down arrows represent downregulation. The red arrows and numbers indicate the upregulated and downregulated gene numbers that overlapped between Ma46 vs. Ma49 and Ma47 vs. Ma48. (**c**) Confirmation of the expression patterns of the *MaUGT* genes involved in coumarin biosynthesis using qRT–PCR. The values shown are the means ± standard deviation of three replicates. *Matubulin* was used as the reference gene. Red lines indicate the expression values (FPKM) from RNA-seq data. CK represents control.

**Figure 4 ijms-22-10826-f004:**
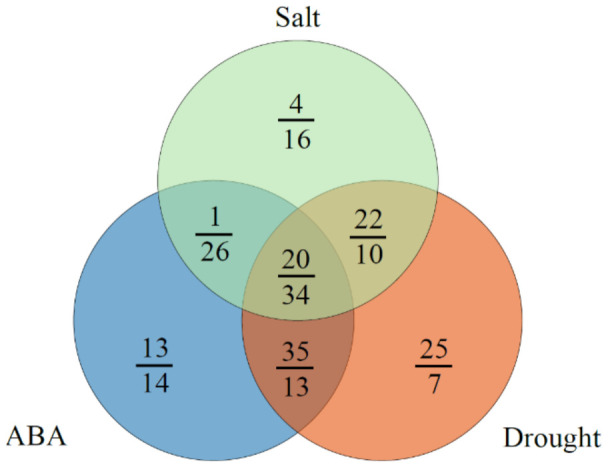
Venn diagram showing the overlap of differentially expressed *MaUGT* genes under ABA and abiotic stress. The letters on the line and below the line indicate the differentially expressed *MaUGT* gene numbers in the shoots and roots, respectively.

**Figure 5 ijms-22-10826-f005:**
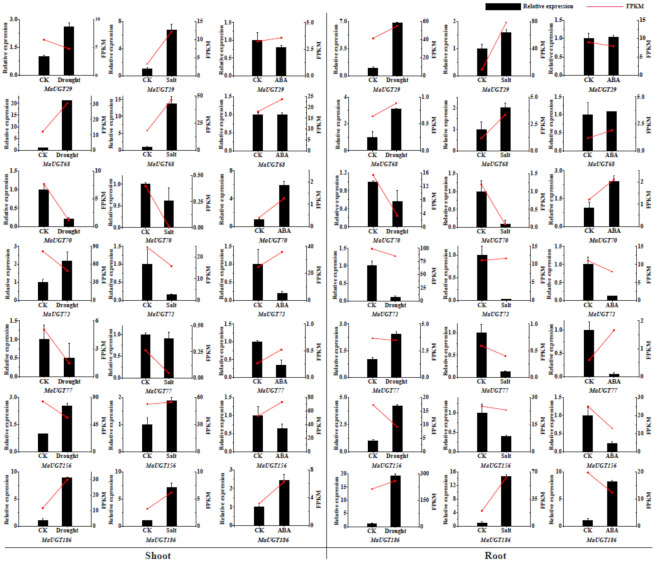
Confirmation of the expression patterns of the *MaUGT* genes under abiotic and ABA stress in shoots and roots using qRT–PCR. The values shown are the means ± standard deviation of three replicates. *Matubulin* was used as the reference gene. Red lines indicate the expression values (FPKM) from RNA-seq data. CK represents control.

**Figure 6 ijms-22-10826-f006:**
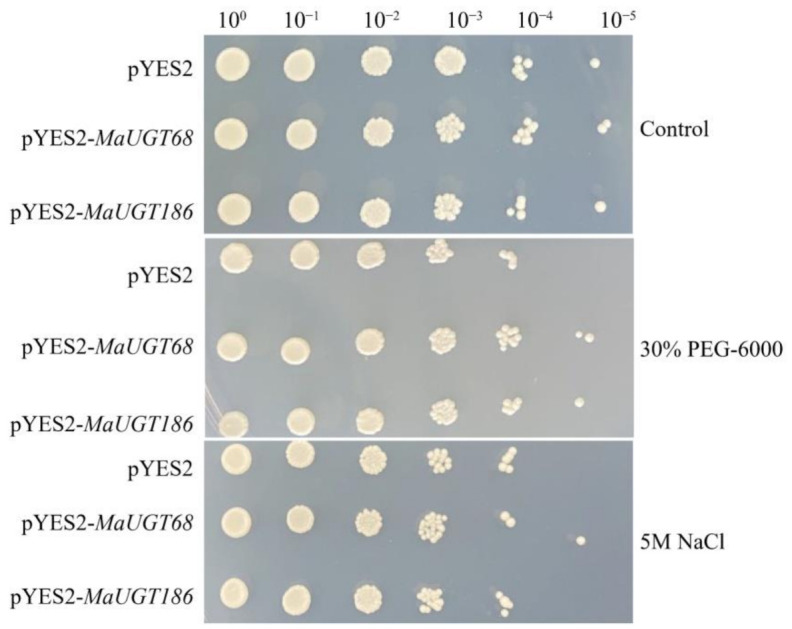
Abiotic stress tolerance analysis of *MaUGT68* and *MaUGT186* in a yeast expression system compared with empty pYES2 (control) yeast. The two yeast cultures were independently grown in synthetic complete (SC)-Ura liquid medium containing 2% (*m*/*v*) galactose at 30 °C for 36 h up to A_600_ = 0.4. Then, the yeast was collected and adjusted with SC-Ura including 2% galactose and cultivated up to A_600_ = 1 for stress analysis. The same number of cells was resuspended in 30% PEG-6000 and 5 M NaCl. Then, serial dilutions (10^0^, 10^1^, 10^2^, 10^3^, 10^4^, 10^5^) were spotted onto SC-Ura agar plates and incubated at 30 °C for 3 days. As a control, yeast with A_600_ = 1 without any stress was also spotted onto SC-Ura agar plates with the same dilutions as the treatments and grown at 30 °C for 3 days.

**Figure 7 ijms-22-10826-f007:**
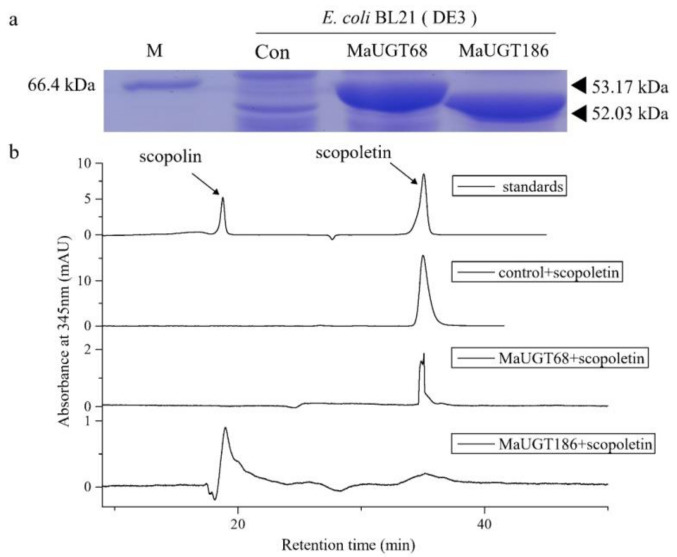
In vitro assay of MaUGT68 and MaUGT186 activity. (**a**). SDS–PAGE analysis of MaUGT68 and MaUGT186 heterologously expressed in *E. coli*. Expression of the proteins in *E. coli* BL21(DE3) was induced by IPTG for 18 h at 28 °C. (**b**). HPLC analysis of the products from the glycosylation assays with MaUGT68 and MaUGT186. Extracts from incubation mixture containing the proteins together with scopoletin and UDP-glucose are compared with authentic scopolin and scopoletin standards. Incubation mixture with empty vector was used as control.

## Data Availability

The genomic data of *M. ablus* are openly available in NCBI (NCBI BioProject ID PRJNA674670).

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
