# Peer review of "Genome-Wide Analysis of the UDP-Glycosyltransferase Family Reveals Its Roles in Coumarin Biosynthesis and Abiotic Stress in Melilotus albus"

_ijms, 2021, doi:10.3390/ijms221910826_

Round 1
Reviewer 1 Report
The manuscript entitled Genome-wide analysis of the UDP-glycosyltransferase family reveals its roles in coumarin biosynthesis and abiotic stress in Melilotus albus authored by Zhen Duan, Qi Yan, Fan Wu, Yimeng Wang, Shengsheng Wang, Xifang Zong, Pei Zhou and Jiyu Zhang has extensively investigated the phylogenetic and structural relationship of UDP-glycosyltransferase genes of Melilotus albus. The authors using transcriptomics also investigated the expression of these genes in various cultivars and tissues under physiological and various stress conditions. To functionally characterize them, the authors chose two genes one up-regulated and one down-regulated and performed yeast survival analysis and biochemical experiments with E.coli extracts containing the two proteins.
The overall flow and the research of the manuscript is well explained and followed.
However, there are certain points that make it inappropriate for publication in IJMS in its present form.
- 2.2 It is not clear how and with which criteria the 17 phylogenetic groups were formed. For example, why N and I are two groups and not one.
- 2.3 “The analysis suggested that MaUGT genes contained 1–4 exons with varying numbers of introns (Supplementary Figure S2). In Sup Fig 2, I can see genes with 0, 1,2 and 3 introns as declared later on in the text. How do the authors calculate the 10 independent intron insertion events? They should elaborate by giving the online tool, the algorithm, the reference…
- 2.3 and figure 2. Authors should explain how the tree originates from figure 1. It was very difficult to understand how the groups were formed. The color code is not the same as in figure 1. Are the two figures related and how?
- 2.3 and figure 2. Why MaUGT26 is colored with magenta and MaUGT162 with orange despite of their close evolutionary relationship? Why MaUGT164 is colored blue and not magenta?
- 2.3 and figure 2. If the thickness of the lines indicates “the length of the amino acid residues”, why the lengths of the lines differ to the proteins? Why are some lines thicker that others?
- 2.3 and figure 2. “the length of the amino acid residues” I think it is syntax error. Do author mean “the length of the amino acid sequences”? or the number of the aa residues ?
- 2.3 and figure 2. Branches between P and G: I think that the beginning of the branch should be black and not red. Authors should check and make sure about the colors.
- 2.3 Supplementary Figure S2. It is not clear from the figure legend how the classification was performed. There are also syntax errors that worsen our understanding (eg. “The subgroup was classified and marked by alternated different; (middle), Protein motif“.). If classification was performed according to motives’ architecture why then different architectures exist within certain groups?
- 2.5 Supplementary Figure S4: “Dendrograms along the top and left sides of the heat map indicate the hierarchical clustering of tissues and genes”. I think that clustering is according to expression of genes and not the genes. I see that clustering and groups on the top have no relation with the trees in figures 1 and 2 . Right ?
- 2.6 NIL should be explained the first time used
- 2.6 for expression experiments tissues where samples are taken should always be reported.
- 2.6 figure 3C: FPKM should be explained in the text and in material and method section. In what does it differ from relative expression?
- 2.6 figure 3: please elaborate in figure legend
- 2.7 Since authors compare NILs (cultivars?) od M.albus in §2.6, they should show their results in §2.7 for the same cultivars. Thus, in Supplementary Figure S7, we cannot figure out the NIL. Moreover. In figure 5 such information is absolutely needed. Are the differences observed for each protein and for each stress condition the same for all NILs? And what does it mean elevated relative expression and less FPKM
- 2.8. “a yeast expression system was selected to analyze the two genes” syntax error.
- 2.8. “However, the transgenic yeast exhibited increased…” transgenic or transformed. From the MM section I understood that the yeast is transformed. Please check
- 2.8. “However, the transgenic yeast exhibited increased survival rates compared with the control yeast under drought and salt treatments (Figure 6)”. I think that this phrase is over-interpretation of the results shown in figure 6. Especially for salt stress and for dilution 10-4 the results are the opposite from what is authors declare. So, please rephrase properly, or provide more experiments
- 2.9. “After incubation of UDP-glucose and scopoletin as substrates together with either the UGT68 or UGT186 protein, the ex-tracts were analyzed by HPLC.“ In MM section the procedure is described differently. Please rephrase appropriately.
- 2.9. figure 7: please elaborate in figure legend. Please change “scopletin” to “scopoletin”
- Discussion: What does “eighteen 71-family UGTs, twenty-eight 72-family UGTs, and six 88-family UGTs, ..”mean? What are the criteria to define these families?
- Discussion: “…indicating the positive roles and difference of MaUGT genes in abiotic stress tolerance”. Please rephrase since the results are over-interpreated
- 4.8 “The treated yeast liquid was diluted to 1:10 and cultured on SC-U/2% (w/v) glucose medium for 2–3 days to observe colony growth, and photos were taken to record the expression of the binding protein”. It should be said that colony growth was tested on agar plates and not on liquid cultures.
- I can see that §2.8, 9, 4.8 and 4.9 do not have a proper scientific language in terms of molecular biology . Please seek for some help.
- There is a discrepancy between numbers of supplementary figures pdf files and the ones embedded with the word file
Minor points
- For all the tools used (MEGA7, TBtools, plantcare, etc.) references should be given
- Paragraph 2.2 “However, Groups R, C, K and F did not have any members” (please add: from albus) if this is what the authors meant. Otherwise, the sentence does not make sense
- “Supplementary Table S3. cis-acting elememts in MaUGT promoters” . Please change to elements
- “Intron gain and loss events, as well as the positions and phases of introns relative to protein sequences, are important cues to understand evolution [29]”. Do authors mean “clues” ?
- “The second highly conserved intron was observed for intron 4, which was predominantly observed in Groups D and L in M. albus. Syntax error”. Please rephrase
- I think that § 4.5 should be after 4.6. please check.
- “Supplementary Figure S6/7” Please elaborate the legend and explain the symbols
Author Response
Thank you for your nice comments on our article. According to your suggestions, we have supplemented two figures and correct several mistakes in our previous draft. Based on your comments, we attached a point-by-point letter to you. The detailed point-by-point responses are listed below.
Special thanks to you for your good comments!
Many thanks and best regards!
Sincerely,
Jiyu Zhang

Reviewer 2 Report
The present manuscript describes (1) the genome-wide identification of UDP-glycosyltransferase-encoding genes in Melilotus albus and (2) attempts of identifying those involved in coumarin biosynthesis, in particular through the use of already described near-isogenic lines (NILs) differing in their coumarin contents and therefore production ability.
Results are properly presented and most of the time well discussed with no over-interpretation. The manuscript is overall rather well written.
Analyses and experiments of the genome-wide analysis, the most important part of the manuscript, are well-conducted with appropriate information, controls and interpretation. The only point which could be added is the number of pseudogenes among MaUGT genes.
In contrast, the part concerning the link(s) with coumarin biosynthesis is more questionable. The biological material is sound and has already been well described in Luo et al. 2017 (DOI:10.1038/s41598-017-04111-y). Nevertheless, once UGT genes differentially expressed between the NILs have been identified (Fig. 3a and 3b), the authors do not clearly establish why they focus on six of them (Fig. 3c) while five are shared between pairwise comparisons in the Venn diagram (Fig. 3b). Similarly, the choice of the seven UGT genes for RT-qPCR validation of differential expression following abiotic stresses (Fig 5) is not explicated (even if six of them are in common with those for differential expression in NILs). UGT genes are usually regulated at the transcriptional level, the overall strategy used by the authors is therefore relevant. However, a clearer crossing of the data from the differential expression between the NILs and the differential expression in abiotic stress conditions should be performed to identify the most relevant candidate genes.
In addition, the last part describing the functional characterization of UGTs MaUGT68 and MaUGT186 (once again why these two?) is not fully convincing. There is discrepancy between the interpretations of the yeast assays: the “results” part concludes on the involvement of both UGTs in the resistance to salt and drought (bottom of page 9) whereas Figure 5 seems to indicate that only MaUGT68 is involved in salt resistance, which the authors do mention in the Discussion part (top of page 12). Moreover, differences seem slight and should be reinforced by proper quantification (Colony forming units? Percentage of the initial inoculum?).
Finally, in vitro characterization of the ability of MaUGTs 68 and 186 to conjugate scopoletin into scopolin should be better described. In particular, the authors do not mention why peak amplitude, albeit starting from the same substrate quantity, is different in the samples with the two recombinant proteins as compared with the empty vector control.
In conclusion, this study is sound, at least for the genome-wide analysis, including the link with coumarin biosynthesis but the functional characterization should be improved. As such, the work cannot be published in IJMS but it deserves resubmission after revisions. Minor revisions to be brought in the text are also listed below.
Minor revisions:
Abstract :
Sentence “which play an important role in plant re- sistance to abiotic stresses“ is redundant with the previous sentence
Introduction:
The authors write “However, the mechanism of glycosylation of coumarins is largely unknown.” There are a few references mentioning UGTs involved in coumarin metabolism such as Chong et al. 2002 (TOGT in tobacco).
Results:
Page 2: chromosome 7 should be added to the chromosomes carrying UGT gene clusters.
Page 6: in Figure 3 caption, the meaning of the red lines should be mentioned.
Page 8: the Figure 5 caption, the meaning of the red lines and of “CK” should be mentioned.
Discussion:
Page 10: there is a wrong mention to the different clades/groups missing in M. albus (letter C is used twice).
Page 11: I think “intron 4” should be replaced by “intron 5” where the “second highly conserved intron” is mentioned.
Materials and Methods:
Paragraph 4.9: the HPLC conditions should be detailed. At least a reference to previous work should be added.
Supplementary Figure 5 is missing from the Supplementary Material Word file.
Author Response
Dear Reviewer:
Thanks for the attention and kind reminding of the referee. According to your suggestions, we have supplemented two figures and correct several mistakes in our previous draft. Based on your comments, we attached a point-by-point letter to you. The detailed point-by-point responses are listed below.
Special thanks to you for your good comments!
Many thanks and best regards!
Sincerely,
Jiyu Zhang

Round 2
Reviewer 2 Report
Following careful reading of the authors' answers and revised version, it appears that all requested points have been carefully addressed. I therefore suggest that the manuscript can be accepted.
For ease of reading, I join the revised manuscript with a few notes included.
